# Meldonium Inhibits Cell Motility and Wound-Healing in Trabecular Meshwork Cells and Scleral Fibroblasts: Possible Applications in Glaucoma

**DOI:** 10.3390/ph16040594

**Published:** 2023-04-15

**Authors:** Cristina Minnelli, Francesco Piva, Monia Cecati, Tatiana Armeni, Giovanna Mobbili, Roberta Galeazzi, Alberto Melecchi, Martina Cristaldi, Roberta Corsaro, Dario Rusciano

**Affiliations:** 1Department of Life and Environmental Sciences, Marche Polytechnic University, 60131 Ancona, Italy; g.mobbili@univpm.it (G.M.); r.galeazzi@univpm.it (R.G.); 2Department of Specialist Clinical Sciences, School of Medicine, Marche Polytechnic University, 60131 Ancona, Italy; f.piva@univpm.it (F.P.); m.cecati@univpm.it (M.C.); t.armeni@univpm.it (T.A.); 3Department of Biology, University of Pisa, 56127 Pisa, Italy; a.melecchi@student.unisi.it; 4Fidia Pharmaceuticals, Research Center, 95123 Catania, Italy; mcristaldi@fidiapharma.it (M.C.); rcorsaro@fidiapharma.it (R.C.); drusciano55@gmail.com (D.R.)

**Keywords:** meldonium (MID), glaucoma, intraocular pressure (IOP)

## Abstract

Meldonium (MID) is a synthetic drug designed to decrease the availability of L-carnitine—a main player in mitochondrial energy generation—thus modulating the cell pathways of energy metabolism. Its clinical effects are mostly evident in blood vessels during ischemic events, when the hyperproduction of endogenous carnitine enhances cell metabolic activities, leading to increased oxidative stress and apoptosis. MID has shown vaso-protective effects in model systems of endothelial dysfunction induced by high glucose or by hypertension. By stimulating the endothelial nitric oxide synthetase (eNOS) via PI3 and Akt kinase, it has shown beneficial effects on the microcirculation and blood perfusion. Elevated intraocular pressure (IOP) and endothelial dysfunction are major risk factors for glaucoma development and progression, and IOP remains the main target for its pharmacological treatment. IOP is maintained through the filtration efficiency of the trabecular meshwork (TM), a porous tissue derived from the neuroectoderm. Therefore, given the effects of MID on blood vessels and endothelial cells, we investigated the effects of the topical instillation of MID eye drops on the IOP of normotensive rats and on the cell metabolism and motility of human TM cells in vitro. Results show a significant dose-dependent decrease in the IOP upon topic treatment and a decrease in TM cell motility in the wound-healing assay, correlating with an enhanced expression of vinculin localized in focal adhesion plaques. Motility inhibition was also evident on scleral fibroblasts in vitro. These results may encourage a further exploration of MID eye drops in glaucoma treatment.

## 1. Introduction

Primary open angle glaucoma (POAG) is a slowly progressing neurodegenerative disease characterized by the death of retinal ganglion cells (RGC) and the retrograde degeneration of the lateral geniculate nucleus and the visual brain cortex [1]. Elevated intraocular pressure (IOP) is considered to be the main risk factor for the development of POAG, although a relevant amount of POAG may develop with normal IOP [2]. In the majority of glaucomatous patients with a normal open angle, there is an increased resistance to the outflow of aqueous humor (AH) from the anterior chamber of the eye, occurring at the level of fluid passage through the trabecular meshwork (TM) and Schlemm’s canal, which is responsible for IOP elevation [3,4]. Although the reason for the decreased permeability of the TM is not yet clear, IOP still remains the only known modifiable risk factor. Therefore, pharmacological treatments for POAG have focused on lowering IOP in the cases of both hypertensive and normotensive glaucoma. This approach has been proven to delay disease progression and decrease the rate of visual field loss [5]. However, not all patients are compliant with the use of a chronic daily use of eye drops [6] and some patients are not fully responsive to the IOP-lowering effect of hypotensive eye drops [7]. In these cases, when a persistent, elevated IOP becomes a clear risk of quick glaucoma progression, a surgical intervention on eye tissues may be advised in order to facilitate AH outflow and decrease the IOP. Laser trabeculoplasty or laser iridotomy are minimally invasive treatments, although with limited efficacy [8]. Depending on the ophthalmologist judgement, a surgical treatment (trabeculectomy) can be proposed. In this case, glaucoma filtering surgery (GFS) with mitomycin C and glaucoma drainage devices remain the standard of surgical care [9]. The aim of filtering surgery is to make a drainage fistula from the anterior chamber to the sub-Tenon space to allow aqueous outflow, thus lowering the IOP. However, a major cause of the failure of this operation is conjunctival fibrosis and scarring in the sub-Tenon’s space, which will reduce AH drainage. The intraoperative application of mitomycin C (MMC) during GFS has increased its success rate by reducing fibrosis soon after surgery. However, despite advances in glaucoma surgical techniques, devices, and post-operative care, fibrosis remains the principal impediment to the long-term control of IOP following surgery and 10% of glaucoma surgeries still fail within the first year [10]. Therefore, the toxicity of MMC, which can be applied only once during surgery, and the overall failure rate of the whole procedure have prompted the research of alternative treatments to maintain the efficiency of the filtration hole [11]. Along this line, we used a molecule (meldonium: Figure 1) with a chemical structure similar to an amino acid, which is used in the treatment of coronary artery disease to improve blood flow by virtue of its vasodilation capacity, thus increasing the oxygen supply to heart tissues [12]. In particular, the anti-ischemic effects of meldonium (MID) may be in part due to the enhancement of NO formation (playing a key role in vasodilation and vascular smooth muscle cell relaxation) by the endothelium [13]. Another mechanism of action of MID is carnitine-dependent and leads to a decrease in the carnitine production and fatty acid accumulation preserving membrane integrity and improving cellular cardiac cells survival during ischemic conditions [14]. Considering the endothelial nature of the TM and the connected Schlemm canal [1], we set out to show whether meldonium administered topically as eye drops to normotensive rats could have an effect on their IOP and whether cells derived from the human TM would respond in vitro to meldonium treatment. The results thus obtained showed a dose–response effect of the IOP to meldonium eye drops and a decrease in the motility of human TM cells, correlating with cytoskeletal changes. The inhibition of motility by meldonium was also evident on scleral fibroblasts in vitro, with interesting implications.

## 2. Results

### 2.1. Effect of MID Eye Drops on IOP in Normotensive Rats: In Vivo Study

At various time points between 15 and 30 min after the instillation of nanomicellar meldonium (MID) eye drops, a dose-dependent decrease in the IOP was observed (Figure 2), which became significant and long-lasting (up until 2 h) at the two higher concentrations (1% and 2% corresponding to 55 and 109 mM, respectively). At the concentration of 1%, a 25% decrease in the IOP was observed, which became a 32% decrease at the 2%concentration. As a point of reference, we verified that the hypotonizing effect of brimonidine (an alpha-agonist) and timolol (a beta-blocker) at their normally used pharmacological concentrations (0.2% for brimonidine and 0.5% for timolol) also achieved, respectively, a 25%–30% reduction in the IOP in the normotensive rat (see Appendix A).

### 2.2. Effect of MID on Cell Viability and Apoptosis Induction of HTMC

In order to investigate the potential cytotoxicity of MID on cells derived from the human TM, cell viability and apoptosis experiments were performed. The effect of aqueous solutions of MID on the primary Human Trabecular Meshwork Cell line (HTMC) cellular viability was determined by the MTT metabolic assay after 24 h and 48 h of treatment with increasing MID concentrations. As shown in Figure 3A, MID induced a slight decrease in cell viability only at the higher concentration tested and only after 48 h of treatment (10% decrease in untreated cells, *p* < 0.05). Results obtained on the HSF cell line confirmed that MID had no relevant cytotoxicity on the ocular cell lines tested (Appendix A).

Next, apoptosis and necrosis were evaluated after 48 h of treatments at the higher MID concentration (109 mM) by the annexin V-FITC/PI staining method. Fluorescence signals were monitored by flow cytometry (Figure 3B,C). No significative differences were observed in the early and late apoptosis levels (*p* > 0.05) nor in the necrosis field.

### 2.3. Effect of MID on Cytoskeleton Organization

In order to uncover whether the MID hypotonizing properties on the IOP could be due to the morpho-physiological effects on HTM cells, we studied the effects of MID on the organization of actin filaments and focal adhesions at the cellular level. F-actin and vinculin were visualized by direct immunostaining (Figure 4A). The pictures show a different distribution of vinculin in MID-treated HTM cells with increased vinculin staining along the border of cell membranes and an increase in focal adhesions with respect to control cells, which flattened MID-treated cells more at the edges (see insets). Actin staining also evidenced a diverse morphology in MID-treated HTM cells.

The quantification of vinculin expression is illustrated by the immunoblot in Figure 4B. A significant increase in vinculin expression was observed after MID treatment (two-fold increase in untreated cells, Figure 4C). On the other hand, the expression of Vimentin, a type III intermediate filament (IF) protein expressed in mesenchymal cells, was not changed by MID treatment (Figure 4B,C), indicating a stability of the fibroblastic phenotype.

### 2.4. MID Prevents Wound Repair in Human Trabecular and Scleral Cell Lines

The observed changes in cytoskeletal organization might suggest an inhibitory effect on cell motility, due to the increase in adhesion plaques. Therefore, in order to investigate the ability of MID to inhibit wound-healing, an established in vitro scratch assay model was used and the results expressed as the percentage of wound closure. Treatments were evaluated after 24 h in the presence of 27 mM and 109 mM MID.

Figure 5A shows the kinetics of wound-healing with HTMC. The wound, in control wells, was already closed after 24 h. With reference to control values set at 100%, MID affected the migratory ability of HTMC with a dose-dependent effect. At 24 h, wound closure was 83% at the lower concentration tested (27 mM) and 65% at the highest concentration tested (109 mM), with relative inhibitions of 20% and 35%.

In order to test whether the inhibitory effect on motility was also evident on other cell types, we checked meldonium effects on a human scleral fibroblast (HSF) cell line. Indeed, the migration inhibitory activity of meldonium was also evident on the HSF, even though the natural motility of these cells was somewhat lower than that of TM cells. In fact, Figure 5B shows that in control wells at 24 h, wound closure was around 50%. However, with reference to control values, MID affected the migratory ability of HSF with a dose-dependent effect. At 24 h, wound closure was 39% at the lower concentration tested (27 mM), and around 26% at the highest concentration tested (109 mM). Hence, the relative inhibition with respect to control was about 24% at 27 mM and 50% at 109 mM, stronger than that observed for HTM.

## 3. Discussion

We have demonstrated in this paper that the vaso-protective drug meldonium (trade name mildronate), normally used to treat cardiovascular diseases, shows hypotonizing properties on the IOP of normotensive rats. The overall result appears to be at least partly caused by the structural organization of the tissue, thus making the TM more permeable to the outflow of the AH. Moreover, an even stronger reduction of cell motility was also exerted on scleral fibroblast cells, with the possible consequence of reducing the fibrotic reaction after filtration surgery.

These results represent a novelty from several points of view. In fact, most of the hypotonizing drugs used in the treatment of POAG function mainly by reducing the influx of AH from the ciliary body into the anterior chamber, such as beta-blockers, carbonic anhydrase inhibitors, and alpha agonists; the latter, together with prostaglandin analogs, also increase the uveoscleral outflow [15]. The TM has been the recent target of a completely different class of hypotonizing drugs, namely the Rho kinase (ROCK) inhibitors. These function by relaxing the structure of the TM and have also been shown to exert some neuroprotective activity by improving blood flow to the optic nerve and increasing ganglion cell survival. They can also act as antifibrotic agents and reduce bleb scarring after glaucoma surgery [16]. Similarly, we had previously shown that forskolin, the active principle contained in the root extract of the plant Coleus forskohlii, was also endowed with hypotonizing activity. The effect on the IOP was due to forskolin stimulating effects on the synthesis of cAMP, which in turn induces the resorption of the AH from the anterior chamber into the ciliary body stroma and indirectly leads to the inhibition of Rho kinase, thus enhancing TM permeability, ultimately resulting in a sensible decrease in the IOP [17]. Along the same lines, we can now demonstrate that meldonium, a molecule already known for the treatment of cardiovascular defects, has the ability to decrease the motility of human trabecular meshwork cells and human scleral fibroblasts, hence with a possible dual effect on the eye of glaucomatous patients undergoing GFS to control IOP. The mechanism by which meldonium appears to exert its effects is at least partly due to its ability to induce cytoskeleton rearrangements, increasing the expression of vinculin, a pivotal protein in the organization of focal adhesion plaques necessary in the regulation of cell migration [18]. In fact, vinculin contributes to the formation and stability of focal adhesions by regulating the generation of contractile stress fibers [19], thereby influencing cell migration speed [20]. Cells deficient in vinculin cannot form lamellipodia, assemble stress fibers, or spread efficiently over a substrate [21]. Most interestingly, vinculin has been shown to be a pivotal protein in the regulation of collective cell migration, a regulatory switch that enables the control of friction and the modulation of cell coupling during migration [22]. This function of vinculin might well correlate with a global change in the architecture of the TM induced by meldonium, making this filtrating tissue more permeable to the AH outflow, hence decreasing the IOP. This mechanism—if confirmed—would be a novel one, also having the cytoskeleton as a target; however, differently to Rho-kinase inhibitors, which work by disassembling the actin cytoskeleton [23], meldonium increases the expression of vinculin and the formation of focal adhesion plaques, thus presumably changing the architecture of the TM tissue.

The inhibition of the cell migration of scleral fibroblasts might have a different meaning in the treatment strategy of hypertensive glaucoma. In fact, the failure of the GFS approach is mostly due to the fibrotic reaction clogging the filtration opening made in the TM tissue, thus jeopardizing its effects on the IOP [24]. Different strategies have been tried in an attempt to prevent or limit this fibrotic response [25]. Promising results were reported with beta-irradiation over trabeculectomy alone, but its comparative long-term efficacy to other current antimetabolites is not yet known [26]. The exploitation of the toxic effects of MMC applied intrasurgically has given good results [24], so that it is now the preferred treatment to attempt the prevention of post-surgical fibrosis. A number of potential approaches to the sustained delivery of antifibrotic agents, such as paclitaxel, sirolimus, and bevacizumab, have appeared promising in preclinical studies [27]. More recently, preclinical studies on rabbits using sustained-delivery films releasing low doses of 5-FU and MMC up to 30 days showed promising results with a reduction in drug toxicity, while preventing fibrosis and preserving bleb function and architecture [28]. However, these methods presently favored to prevent the fibrotic reaction and are based on the toxic effect exerted by the treatment, can be applied only once, intrasurgically, and therefore have limited efficacy.

The effects of meldonium eye drops on the IOP and on cell motility are novel and may provide a new treatment strategy after GFS. The main meldonium mechanism of action is the inhibition of l-carnitine biosynthesis and the increase in peroxisome activity in the cytosol. In fact, Mildronate™ (the commercial form of meldonium) was originally designed to inhibit carnitine biosynthesis in order to prevent and treat the accumulation of intermediate cytotoxic products of fatty acid beta-oxidation in ischemic tissues and to block this highly oxygen-consuming process [29]. Alternatively, meldonium may work via the stimulation of nitric oxide production in the vascular endothelium through a modification of the γ-butyrobetaine and γ-butyrobetaine ester pools, thus inducing vaso-relaxation and improving the blood flow [13]. Meldonium-treated patients demonstrated a marked reduction in systolic BP and heart rate during pronounced summer heat and an improved quality of life [30]. Therefore, Mildronate™ is used in the treatment of heart ischemia and its consequences. Because of its adaptogen effects, meldonium is used by athletes with the purpose of increasing recovery rate or exercise performance [31], and therefore, it has been included on the list of doping agents [32]. However, doping doses of meldonium to attain measurable results are in the range of grams per day, taken orally, and its excretion in urine is diluted over several months [33]. Therefore, it is not expected that the administration of a few milligrams as eye drops would result in a relevant doping effect or become detectable in urine, despite chronic treatment. Nonetheless, this must be proven by appropriate analytical methods. The use of meldonium formulated as nanomicellar eye drops as presented in this study would have the double advantage of exerting a hypotonizing action per se and of reducing the cell motility of HTM cells and HSF, thus exerting a hypothetical anti-fibrotic action, prolonging the effects of GFS. The formulation in nanomicelles favors the penetration of the eye drops through the ocular surface barrier, reaching the target tissues in the anterior eye segment, as we had already demonstrated with the nanomicellar formulation of melatonin eye drops [34]. Moreover, since meldonium does not show any toxicity signs after application on the ocular surface of experimental animals, the topically applied eye drops can be used for a long time after GFS, allowing a constant and tight control of the fibrotic reaction.

Considering that in vitro and in vivo studies support the important role of endogenous NO in regulating IOP homeostasis [35], the ability of MID to lower IOP could be in part ascribed to this mechanism of action. However, since L-carnitine is involved in mitochondrial metabolism, the inhibition of carnitine transporter OCTN2, expressed by ocular epithelial cells, may slow down the metabolism of trabecular cells and scleral fibroblasts, resulting in the apparent toxicity and inhibition of cell movement. So far, we cannot say how much of the IOP-decreasing effect of MID is due to the enhancement of NO production and the effect of vasodilation and how much is possibly due to the depression of carnitine production and the inhibition of cell motility, likely leading to a morphological rearrangement of the trabecular meshwork.

Finally, the most significant limit of this study is its preliminary nature; therefore, the supposed anti-fibrotic effect of meldonium must be studied and confirmed in an in vivo setting in order to show the real efficacy of the topical treatment and its absence of long-term toxicity on other ocular structures. Nonetheless, these data encourage such further studies, with the hope of finding a better and more definitive treatment, giving GFS a more consistent role in the management of hypertensive glaucoma.

## 4. Materials and Methods

### 4.1. Materials

3-(4,5-dimethylthiazol-2-yl)-2,5-diphenyltetrazolium bromide (MTT) was purchased by Sigma-Aldrich (Milan, Italy). Fibroblast culture medium (P60108) was purchased from Innoprot (48160 Derio, Bizkaia, Spain). FITC Annexin V Apoptosis Detection Kit was purchased from Biolegend (San Diego, CA, USA). Others cell culture reagents were obtained from Euroclone (Milan, Italy).

### 4.2. Animal Treatment with Meldonium Eye Drops

The formulation of meldonium dihydrate (Acef S.p.A., Piacenza, Italy) in nanomicelles (Soluplus^®^, 115.000 g/mol; BASF, Ludwigshafen, Germany) was prepared by the direct dissolution method. Soluplus (11.5% *w*/*v*) was dispersed in Tris buffer at pH 7.4. Meldonium dihydrate was then added to the solution, and the system was maintained under magnetic stirring at room temperature for 24 h in order to allow complete dispersion. The formulation was sterilized by filtration through 0.22 μm sterile membranes (Minisart, Sartorius, Gottingen, Germany). The pH was measured using a calibrated pH-meter (XS instrument, Modena, Italy). Osmolarity was measured by an osmometer (Osmomat 3000; Gonotec, Berlin, Germany). The mean particle size (Z-Ave) and the polydispersion index (PDI) were determined using a Nano-Sizer ZS90 (Malvern Instruments, Malvern, UK). Samples were diluted ten-fold with water before analysis. Soluplus^®^ micelles had an average size of 66.1 ± 0.06 nm and a polydispersion index (PDI) 0.083 ± 0.045. The nanosuspension was transparent and slightly opalescent as compared to water.

Animals were used in agreement with the Association for Research in Vision and Ophthalmology’s statement on the use of animals in ophthalmic and vision research. The study also conforms to the European Communities Council Directive (2010/63/UE) and the Italian guidelines for animal care (DL 26/14). The experimental protocol was approved by the Commission for Animal Wellbeing of the University of Pisa (protocol n. 133/2019-PR). Rats (Sprague Dawley strain, 200 g body weight, 2–3 months of age) were obtained from Charles River Laboratories Italy (Calco, Italy). In order to make tonometry less stressful, the rats were acclimatized for 1 week. To evaluate the effects of meldonium eye drops (given at 27, 55 and 109 mM) on normotensive rats, 3 rats (6 eyes, 3 measurements/eye/time point) were used for each formulation. Rats were treated with 10 µL per eye of the meldonium formulations prepared as above. The IOP was measured from time 0 (before administration) to 6 h after administration using the TonoLab device (iCare, Vantaa, Finland). The animals did not show any algic or inflammatory reactions after the use of eye drops. Control animals received the vehicle only.

### 4.3. Cell Culture

The primary human trabecular meshwork cell line (HTMC: P10879) and the related fibroblast culture medium (P60108) were both purchased from Innoprot (48160 Derio, Bizkaia, Spain). The culture medium was supplemented with 2% FCS and antibiotics (1% pen-strep). Experiments were performed using HTMC no older than the 5th passage. Human scleral fibroblasts (HSF) were purchased from Lifeline Cell Technology (Frederick, MD, USA) and cultivated in Lifeline FibroLife medium supplemented with the FibroLife S2 LifeFactors Kit (ibid., Cat. no. LL-0011). HSF were grown at 37 °C in a humidified atmosphere containing 5% CO_2_.

### 4.4. Cellular Metabolic Activity—MTT Assay

The estimation of cell viability was based on the number of metabolically active cells, as determined by the 3-(4,5-dimethylthiazol-2-yl)-2,5-diphenyltetrazolium bromide (MTT) assay as previously described [36]. Briefly, HTMC were seeded in 24-well plates at 5 × 10^3^ cells/cm^2^ to reach 80% of confluence the next day; then, medium was removed and replaced with 0.5 mL of complete culture medium supplemented with increasing concentrations of MID (3.5, 7, 14, 27, 54, and 109 mM). After 48 h of incubation, the medium from each well was removed and replaced with fresh medium supplemented with MTT at a final concentration of 0.1 mg/mL; samples were incubated for 3 h at 37 °C in a 5% CO_2_ atmosphere until formazan crystals were formed. Next, 0.4 mL of DMSO were added to each well and mixed thoroughly by shaking to solubilize the MTT formazan crystals. Absorbance was read on a multiwell scanning microplate reader (BioTek Synergy HT MicroPlate Reader Spectrophotometer from Agilent, Santa Clara, CA, USA) at 570 nm using the extraction buffer as a blank. The optical density in the control group (untreated cells) was considered to have 100% viability. The relative cell metabolic activity (%) was calculated as (A570 of treated samples/A570 of untreated samples) × 100. Determinations were carried out in triplicate in each experiment and the mean ± SD from three independent experiments was calculated.

### 4.5. Cell Migration Analyses

Cell migration was evaluated by the scratch wound assay, as previously described [37]. Briefly, HTMC or HSF cells were seeded on 6-well plates at 7 × 10^3^ cells/cm^2^ in their respective complete media supplemented with 2% FCS. After 24 h, scratch wounds were created with 1000 μL pipette tips on the pre-seeded confluence cells. The culture was washed with PBS after scratch wound induction and replaced by complete medium (2% FCS) containing MID at final concentrations of 27 and 109 mM. Photomicrographs were taken at time 0 (immediately following the scratch wound) and after 24 h. The wound gaps were measured by ImageJ (version 1.47; NIH, Bethesda, MD, USA). The migration percentage was calculated by the average area reduction at 24 h and compared to time 0. The experiments were performed in triplicate, and the means ± SD from the three independent experiments was calculated.

### 4.6. Measurement of Apoptosis by Annexin V Staining

The Annexin V Thermo Fisher (Waltham, MA, USA) assay kit was used (Biolegend, San Diego, CA, USA), according to the manufacturer’s instructions. HTMC were seeded on 6-well plates at a density of 5 × 10^3^ cells/cm^2^. After 24 h, cells were exposed to MID (109 mM) for 48 h; then, the cells were trypsinized, collected, and centrifuged. After washing twice with cold PBS, untreated and MID-treated cells were resuspended in 1X Binding Buffer at a final concentration of 1.0 × 10^6^ cell/mL. Annexin V-FITC (0.25 mg/mL) and Propidium Iodide (PI) (1 mg/mL) were added to the cell suspension and incubated for 10 min at room temperature, avoiding direct light exposure. Samples were analyzed using the Beckman Coulter EPICS XL flow cytometer at an excitation wavelength of 488 nm. Annexin V-FITC was detected as a green fluorescence (FL1 channel) and propidium iodide (FL-3 channel) was detected as a red fluorescence. Early apoptosis was defined by Annexin V+/PI- staining, late apoptosis was defined by Annexin V+/PI+ staining, and necrosis was defined by Annexin V−/P+ staining. The fluorescence intensity was recorded on an average of 4000 cells from each sample.

### 4.7. Immunofluorescence

Cells were cultured on poly-D-lysine–coated imaging chambers with a coverslip bottom (µ-Slide 8-well high Glass Bottom #80827 from IBIDI GmbH, Gräfelfing, Germany) and cells were fixed in 4% paraformaldehyde in PBS, permeabilized for 10 min in 0.1% Triton X-100 in PBS, and blocked (1% BSA) for 1 h at RT, followed by incubation (4 °C) with anti-vinculin mouse monoclonal antibody (#MA511690; Invitrogen; 1:700; Waltham, MA, USA) for 3 h. Cells were washed 3 times in PBS and incubated with goat anti-mouse secondary antibody conjugated with Alexa Fluor 647 for 1 h at RT (#A21236, Thermo Fisher, 1:1000, Waltham, MA, USA). Next, cells were incubated with Alexa Fluor 488–phalloidin (#A12379, 1:400; Thermo Fisher, Waltham, MA, USA) and Hoechst 33,342 (#H3570, Waltham, MA, USA Invitrogen) for 30 min. The images were taken by using the Eclipse Ti2E microscope (Nikon, Tokyo, Japan).

### 4.8. Immunoblot

Cells were lysed with RIPA-buffer (Tris-HCl 50 mM pH 8.0, NaCl 150 mM, EDTA 2 mM, Triton X-100 0.5%) containing protease inhibitor (#P8340 Sigma-Aldrich, Milan, Italy) on ice for 40 min and centrifugated at 12000× *g* for 10 min at 4 °C. Supernatants were collected and protein concentration was determined by the Bradford protein assay (#5000205; Bio Rad; Segrate, Milan, Italy). Equal amounts of denatured proteins (20 μg) were separated on a 4–20% precast SDS-PAGE system (EC6021BOX, ThermoFisher, Waltham, MA, USA) and then transferred onto immunoblot PVDF membranes (#1620177; Bio Rad; Segrate, Milan, Italy). The membrane was blocked with Everyblot blocking buffer (#12010020, Bio Rad; Segrate, Milan, Italy) at 25 °C for 5 min. The blots were incubated with mouse monoclonal antibodies against vinculin (MA5-11690; 1:300; Invitrogen; Waltham, MA, USA) and vimentin (#3932S; 1:2000; Cell Signaling; Danvers, MA, USA) at room temperature for 1 h. All membranes were washed three times with 1× Tris-Buffered Saline, 0.1% Tween20 Detergent (TBST) for 10 min each time and incubated with anti-mouse horseradish peroxidase (HRP)-conjugated secondary antibodies, in accordance with the manufacturer’s instructions, at room temperature for 1 h. As internal control for protein loading, membranes were stripped and probed again with rabbit monoclonal GAPDH antibodies (#97166; 1:1000; Cell Signaling; Danvers, MA, USA) for 1 h. Membranes were developed with enhanced SuperSignal™ West Pico PLUS Chemiluminescent Substrate (#34580; Thermo Fisher Scientific, Waltham, MA, USA). The chemiluminescent signal of proteins detected in blots was acquired using the ChemiDoc XRS+ System (Bio-Rad Laboratories, Hercules, CA, USA) and analyzed using ImageJ software (version 1.47; NIH, Bethesda, MD, USA).

### 4.9. Statistical Analyses

Data are presented as means ± S.D. (standard deviations). The statistical comparison of differences among groups of data was carried out using a one-way analysis of variance (ANOVA), followed by Tukey’s post hoc test using GraphPad Prism (San Diego, CA, USA). Values of *p* < 0.05 were considered statistically significant, and values of *p* < 0.01 were considered highly significant.

## 5. Conclusions

In conclusion, here, we demonstrated the properties of an old molecule (meldonium) on a novel application in the field of glaucoma treatment. We took advantage of mechanisms of action so far unexploited in glaucoma therapy, i.e., carnitine metabolism and the cytoskeleton, finally leading to a sensible decrease in the IOP and the likely prevention of the fibrotic reaction after GFS. However, we cannot exclude that the IOP decreasing effect of MID could be also due to the enhancement of NO production and the effect of vasodilation. Therefore, the use of meldonium as eye drops in a trans-epithelial formulation with nanomicelles could represent a novel conservative treatment after GFS, prolonging the hypotonizing effects of the surgical intervention and adding a further relevant hypotonizing effect. The low concentration of meldonium required in the topical application (in the range of milligrams), the scarce penetration expected in the blood stream, and its slow renal excretion should not cause doping effects in treated patients and would likely be undetectable in their urine.

## 6. Patents

The following patent was obtained: IT201900002441A1.

## Figures and Tables

**Figure 1 pharmaceuticals-16-00594-f001:**
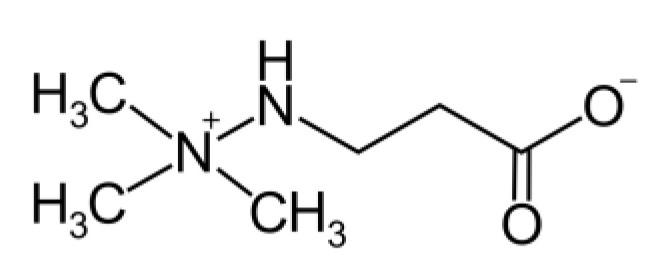
The chemical structure of meldonium (trade name mildronate).

**Figure 2 pharmaceuticals-16-00594-f002:**
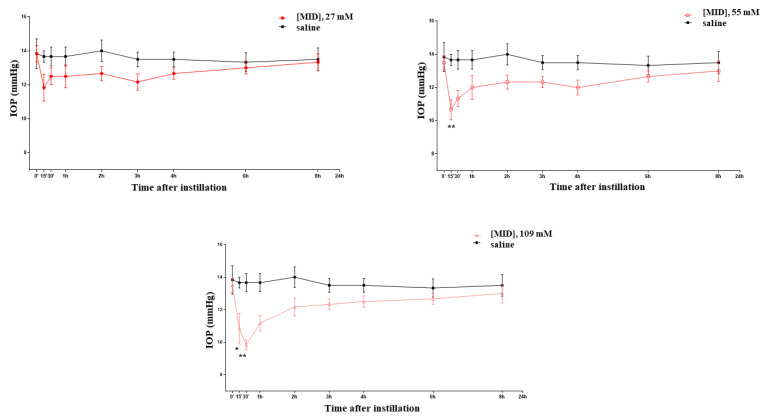
Effects on the IOP of meldonium (MID) eye drops at increasing concentrations. * *p* < 0.05; ** *p* < 0.01.

**Figure 3 pharmaceuticals-16-00594-f003:**
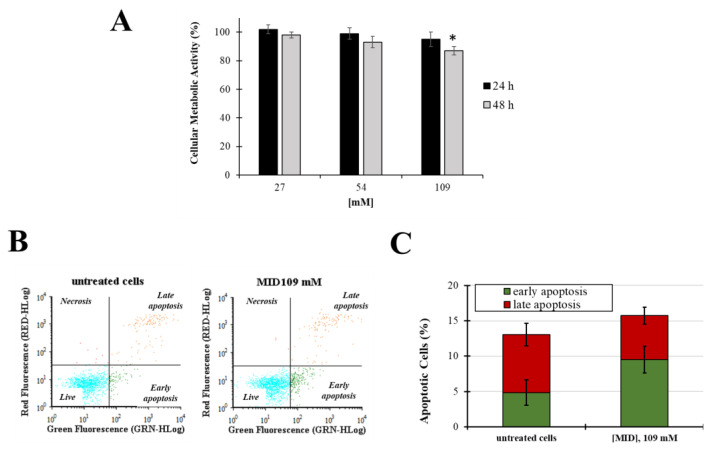
(**A**) Cellular metabolic activity in HTMC treated with increasing concentrations of MID for 48 h in complete culture medium (2% FCS); cell viability was determined by MTT assay. Data are expressed as the means ± S.D. of three independent experiments, each performed in triplicate. * *p* < 0.05 with respect to untreated control cells. (**B**) Representative flow cytogram of propidium iodide (*y*-axis) vs. annexin V-FITC (*x*-axis). (**C**) Histogram of the total percentage of apoptotic cells, in which values represent the means of three experiments. * *p* < 0.05 vs. untreated cells.

**Figure 4 pharmaceuticals-16-00594-f004:**
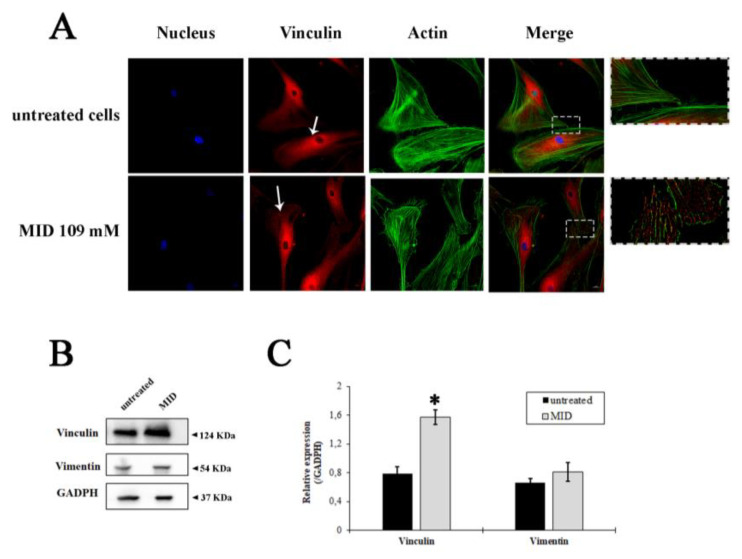
Fluorescence staining of HTMC with FITC-phalloidin (green), Vinculin antibody (red) and Hoechst (blue) after 48 h of treatments with MID. (**A**) Representative fluorescence microscopy images showing cytoskeletal changes characterized by F-actin and vinculin cell distribution. (**B**) Western blotting showing the expression change of vinculin and vimentin in response to MID treatment. The samples derive from the same experiment; gels and blots were processed in parallel and reported in SI. Each blot was cut in order to isolate the individual proteins and probed with specific antibodies, as illustrated in M&M. (**C**) Histogram showing the quantification of the change in vinculin and vimentin expression with respect to GAPDH, used as an internal loading control. * *p* < 0.05.

**Figure 5 pharmaceuticals-16-00594-f005:**
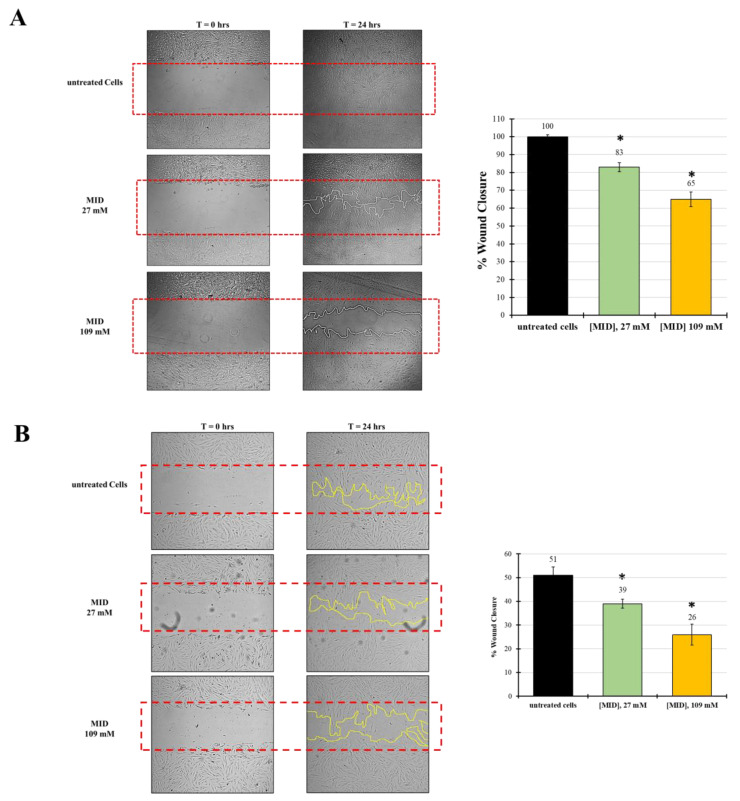
The effect of MID on (**A**) HTMC or (**B**) HSF cell migration was evaluated by the scratch wound assay. Cells were treated with MID at selected concentrations; images were taken at time 0 and 24 h after the artificial wound was made. The red dotted box represents the size of the original wound. The wound area was measured by Image J software. The percentage of migration was calculated as the ratio of the average area reduction at the time of analysis to time 0. Data represent the mean of triplicated experiments ± standard deviation. Scale bar: 100 μm. * *p* < 0.05 significative differences with respect to control wells (ANOVA).

## Data Availability

Data is contained within the article and supplementary material.

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
