# Peer review of "Meldonium Inhibits Cell Motility and Wound-Healing in Trabecular Meshwork Cells and Scleral Fibroblasts: Possible Applications in Glaucoma"

_pharmaceuticals, 2023, doi:10.3390/ph16040594_

Round 1
Reviewer 1 Report
1. The article has a rigid structure and clear logic, and it explores the effects of Meldonium on the eyes. Some hypotheses are put forward, and can experiments be conducted to verify some of them in advance?
2. Meldonium is a cardiovascular drug, and the eyes are an organ with a dense distribution of blood vessels. How the drug affects the blood vessels needs to be analyzed, and do all cardiovascular drugs can help treat eye diseases?
Author Response
Comments and Suggestions for Authors
- The article has a rigid structure and clear logic, and it explores the effects of Meldonium on the eyes. Some hypotheses are put forward, and can experiments be conducted to verify some of them in advance?
Of course, more experiments are necessary to confirm the hypotheses proposed in this paper, and we are programming to run them in the months to come. So far, we believe that it is worthwhile to share our findings with the scientific community, to encourage more research on this topic.
- Meldonium is a cardiovascular drug, and the eyes are an organ with a dense distribution of blood vessels. How the drug affects the blood vessels needs to be analyzed, and do all cardiovascular drugs can help treat eye diseases?
Thank you for your comment. We described the main mechanisms of action of meldonium as anti-ischemic drug in the Introduction by the following sentence “In particular, the anti-ischemic effects of meldonium (MID) may be in part due to the enhancement of NO formation (having a key role in vasodilation and vascular smooth muscle cell relaxation) by the endothelium [13]. Another mechanism of action of MID is carnitine-dependent and leads to a decrease of carnitine production and fatty acid accumulation, thus preserving membrane integrity and improving cellular cardiac cells survival during ischemic conditions [14]”.
Moreover, since converging evidence support the important role of endogenous NO in regulating IOP homeostasis, cardiovascular drugs that act by increasing vasodilation and NO levels could be potentially effective in some eye diseases such as ocular hypertensive glaucoma. Nitroglycerin, a drug commonly used for the treatment of angina [Moncada S Higgs A Furchgott R. International Union of Pharmacology Nomenclature in Nitric Oxide Research. Pharmacol Rev . 1997; 49: 137–142; Marsh N, Marsh A. A short history of nitroglycerine and nitric oxide in pharmacology and physiology. Clin Exp Pharmacol Physiol. 2000;27:313–319.], has been shown to lower the IOP in vivo both in animals and humans [Wizemann AJ Wizemann V. Organic nitrate therapy in glaucoma. Am J Ophthalmol . 1980; 90: 106–109; Schuman JS Erickson K Nathanson JA. Nitrovasodilator effects on intraocular pressure and outflow facility in monkeys. Exp Eye Res . 1994; 58: 99–105; Nathanson JA. Nitrovasodilators as a new class of ocular hypotensive agents. J Pharmacol Exp Ther . 1992; 260: 956–965.]. In this context, we added in the Discussion section the following sentence “Considering that in vitro and in vivo studies support the important role of endogenous NO in regulating IOP homeostasis [34], the ability of MID to lower IOP could be in part ascribed to this mechanism of action. However, since L-carnitine is involved in mitochondrial metabolism, the inhibition of carnitine transporter OCTN2, expressed by ocular epithelial cells, may slow down the metabolism of trabecular cells and scleral fibroblasts resulting in apparent toxicity and inhibition of cell movement. So far, we cannot say how much of the IOP decreasing effect of MID is due to the enhancement of NO production and the effect of vasodilation, and how much it is possibly due to the depression of carnitine production and the inhibition of cell motility, likely leading to a morphological rearrangement of the trabecular meshwork.”
Reviewer 2 Report
The article titled ¨ Meldonium inhibits cell motility and wound healing in trabecular meshwork cells and scleral fibroblasts: possible applications in glaucoma¨. The main topic is interesting because it explores the application of meldonium as a possible treatment for glaucoma. Nevertheless, some issues need clarification or modification.
1. The results sections start with the in vivo assay. Nevertheless, this corresponds to the last method declared in the Material and Methods section, thus confusing the reader. Do the authors execute all experiments with nano micelles eye drops? Please clarify, and modify the order of the result or methods sections. Even the discussion section has the order of in vitro and, finally, in vivo approach. This is incossinting.
2. In lines 100-101, please declare the results obtained in HSF cells.
3. In line 246, the authors mentioned that MID is listed in the doping agents associated o an athlete's recovery rate or exercise performance. The question is: how do authors propose the MID for treating glaucoma if the MID is listed in the doping agents? Please expand this argument.
4. Please, clarify the reason for using the 109mM concentration for in vitro assays, considering that 10% of cell viability reduction in the in vitro assay.
5. Do the authors include an appropriate positive and negative control in the in vitro and in vivo assays? Please clarify in every experiment.
6. The declaration in Lines 385-387 is part of the manuscripts?. Please, eliminate if not.
7. Please clarify the data available statement. Are the data available or not?
Author Response
Comments and Suggestions for Authors
The article titled ¨ Meldonium inhibits cell motility and wound healing in trabecular meshwork cells and scleral fibroblasts: possible applications in glaucoma¨. The main topic is interesting because it explores the application of meldonium as a possible treatment for glaucoma. Nevertheless, some issues need clarification or modification.
- The results sections start with the in vivo assay. Nevertheless, this corresponds to the last method declared in the Material and Methods section, thus confusing the reader. Do the authors execute all experiments with nano micelles eye drops? Please clarify, and modify the order of the result or methods sections. Even the discussion section has the order of in vitro and, finally, in vivo approach. This is inconsistent.
The in vivo experiments have been moved before the in vitro assays in the methods section. The nanomicellar eye drops were used only for the in vivo experiments because their use is important to favor the penetration of the eye drops through the ocular surface barrier, reaching the target tissues in the anterior eye segment (Line 109, 104-105, 257-260).
- In lines 100-101, please declare the results obtained in HSF cells.
We added the cell viability data obtained by MTT assay in the supplementary materials (Figure S2). Overall, MID treatment did not affect the cell viability of HSF cells (Line 108-109).
- In line 246, the authors mentioned that MID is listed in the doping agents associated o an athlete's recovery rate or exercise performance. The question is: how do authors propose the MID for treating glaucoma if the MID is listed in the doping agents? Please expand this argument.
We added the following sentence in the Discussion section “However, doping doses to attain measurable results are in the range of grams per day, taken orally, and its excretion in urine is diluted along several months [33]. Therefore, it is not expected that the administration of few milligrams as eye drops should result in a relevant doping effect, and is detectable in urines, despite a chronic treatment. Nonetheless, this must be proved by appropriate analytical methods” (Line 255).
- Please, clarify the reason for using the 109mM concentration for in vitro assays, considering that 10% of cell viability reduction in the in vitro assay.
Considering that among the effects of meldonium, there is a depression of the mitochondrial activity, that impinges on the MTT assay, the decrease in vitality shown by the assay could be due not to cell death, but to a metabolic decrease (DOI: 10.1177/2397847320915143). In fact, since L-carnitine is involved in mitochondrial metabolism, the inhibition of carnitine transporter OCTN2, expressed by ocular epithelial cells, may slow down the metabolism of trabecular cells and scleral fibroblasts resulting in apparent toxicity and inhibition of cell movement (this sentence is now added at line 271 in the Discussion section). Besides, a 10% decrease either due to metabolism or to cell death, is an acceptable toll to pay for the results, also compared to the high toxic effects of mitomycin C, which is presently used during filtration surgery.
- Do the authors include an appropriate positive and negative control in the in vitro and in vivo assays? Please clarify in every experiment.
Negative controls have always been included in all experiments, and are represented by treatment with vehicle alone, showing no effects as compared to untreated control cells. Positive controls are normally requested to show that the biological system is responsive to the given stimulations. In this study, all tested cells were normally responsive to meldonium treatment, so that we reputed unnecessary to use a positive control. Moreover, the precise molecular mechanisms by which meldonium is exerting its effects on cell motility has not been explored in depth, so that it would be difficult to choose an appropriate positive control. This will be the matter of further studies. We have added in the results section a reference to the response of rats to the treatment with well-known hypotonizing eye drops (a beta blocker and an alpha-agonist) as a term of reference for meldonium effects: “As a term of reference, we have verified that the hypotonizing effect of brimonidine (an alpha-agonist) and timolol (a beta-blocker) at their normally used pharmacological concentrations (0.2% for brimonidine and 0.5% for timolol) also achieves respectively a 25% - 30% reduction of the IOP in the normotensive rat (see supplementary, Figure S1)”.
- The declaration in Lines 385-387 is part of the manuscript? Please, eliminate if not.
We removed the sentence.
- Please clarify the data available statement. Are the data available or not?
The data presented in this study are available on request. Data are also reported in the Supplementary Material. The data are not publicly available because proprietary of Fidia Pharmaceuticals.
Reviewer 3 Report
In this paper Minnelli et al. studied the effects of topical instillation of Meldonium, a well known drug with established beneficial effects in cardiovascular, neurological and metabolic diseases due to its anti-ischaemic and cardioprotective properties, on the intraocular pressure of normotensive rats in vivo, and on the cell metabolism and motility of human trabecular meshwork cells in vitro. The study displays interesting results. I appreciate the work.
Minor comments:
1. On what criteria was based the choice of concentrations of MID in this study?
2. Statistical analysis should be shown in the materials and methods section.
3. Figure 5 (p. 6) is mistakenly shown as figure 3.
4. "Conclusions" section that highlights the main finding of the study is better to be added.
Author Response
Comments and Suggestions for Authors
In this paper Minnelli et al. studied the effects of topical instillation of Meldonium, a well-known drug with established beneficial effects in cardiovascular, neurological and metabolic diseases due to its anti-ischemic and cardioprotective properties, on the intraocular pressure of normotensive rats in vivo, and on the cell metabolism and motility of human trabecular meshwork cells in vitro. The study displays interesting results. I appreciate the work.
Minor comments:
- On what criteria was based the choice of concentrations of MID in this study?
Considering the cell viability results on both HTM and HSF cells, we decided to use the higher MID concentration (109 mM). The 10% decrease in cell viability shown in HTM cells could be due not to cell death, but to a metabolic decrease (DOI: 10.1177/2397847320915143). In fact, since L-carnitine is involved in mitochondrial metabolism, the inhibition of carnitine transporter OCTN2, expressed by ocular epithelial cells, may slow down the metabolism of trabecular cells and scleral fibroblasts resulting in apparent toxicity and inhibition of cell movement (this sentence is now added at line 271 in the Discussion section). Besides, a 10% decrease either due to metabolism or to cell death, is an acceptable toll to pay for the results, also compared to the high toxic effects of mitomycin C, which is presently used during filtration surgery.
- Statistical analysis should be shown in the materials and methods section.
We added Statistical analyses in the materials and methods section.
- Figure 5 (p. 6) is mistakenly shown as figure 3.
Yes. It was a mistake. The figure number was now corrected.
- "Conclusions" section that highlights the main finding of the study is better to be added.
We added the conclusion. “In conclusion, we have shown here the properties of an old molecule (meldonium) on a novel application in the field of glaucoma treatment. We took advantage of mechanisms of action so far unexploited in glaucoma therapy, i.e., carnitine metabolism and the cytoskeleton, finally leading to a sensible decrease of the IOP and the likely prevention of the fibrotic reaction after GFS. However, we cannot exclude that the IOP decreasing effect of MID could be also due to the enhancement of NO production and the effect of vasodilation. Therefore, the use of meldonium as eye drops in a trans-epithelial formulation with nanomicelles could represent a novel conservative treatment after GFS, prolonging the hypotonizing effects of the surgical intervention, and adding on itself a relevant hypotonizing effect. The low concentration of meldonium requested in the topical application (in the range of milligrams), the scarce penetration expected in the blood stream and its slow renal excretion, should not exert doping effects in treated patients, and likely be undetectable in their urine.”
Round 2
Reviewer 1 Report
I think it is acceptable now.
Reviewer 2 Report
The author's responses to all reviewer concerns. Now the document is improved and ready to be published.